# SST-ResNet: A Sequence and Structure Information Integration Model for Protein Property Prediction

**DOI:** 10.3390/ijms26062783

**Published:** 2025-03-19

**Authors:** Guowei Zhou, Yanpeng Zhao, Song He, Xiaochen Bo

**Affiliations:** 1Academy of Medical Engineering and Translational Medicine, Tianjin University, Tianjin 300072, China; 2022235039@tju.edu.cn; 2Academy of Military Medical Sciences, Beijing 100850, China; zyp182531903@163.com; 3School of Medicine, Shanghai University, Shanghai 200444, China

**Keywords:** drug discovery, deep learning, protein property, multimodal model, structure token, information integration

## Abstract

Proteins are the basic building blocks of life and perform fundamental functions in biology. Predicting protein properties based on amino acid sequences and 3D structures has become a key approach to accelerating drug development. In this study, we propose a novel sequence- and structure-based framework, SST-ResNet, which consists of the multimodal language model ProSST and a multi-scale information integration module. This framework is designed to deeply explore the latent relationships between protein sequences and structures, thereby achieving superior synergistic prediction performance. Our method outperforms previous joint prediction models on Enzyme Commission (EC) numbers and Gene Ontology (GO) tasks. Furthermore, we demonstrate the necessity of multi-scale information integration for these two types of data and illustrate its exceptional performance on key tasks. We anticipate that this framework can be extended to a broader range of protein property prediction problems, ultimately facilitating drug development.

## 1. Introduction

Proteins are the major macromolecules carrying out the essential functions in biology. They are composed of sequences of amino acids (AAs) joined by peptide bonds. During biosynthesis on the ribosome, a natural protein folds into its tertiary (3D) structure to carry out its function. With the rapid advancements in low-cost sequencing technologies, a vast array of protein sequences has been identified in recent years. However, the property annotation of newly discovered protein sequences remains a resource-intensive and time-consuming process, highlighting the urgent need for accurate and efficient methods to bridge the gap between protein sequences and their properties.

With the continuous accumulation of data and advancements in deep learning methodologies, leveraging deep learning approaches for protein property prediction has emerged as the optimal solution to address this challenge. Given that the properties of many proteins are intrinsically linked to their folded structures, numerous data-driven methods focus on deriving representations based on these protein structures. Recent breakthroughs in deep learning approaches, including AlphaFold [1,2,3] and trRosetta [4], have been widely adopted for accurately predicting protein structures. Phenotypes are influenced by more than just genotype, and similarly, protein sequences do not always fold into identical three-dimensional structures under varying physiological conditions. Proteins such as G-protein-coupled receptors (GPCRs) [5] and hemoglobin also undergo changes in response to the presence or absence of allosteric modulators, which directly influence their properties, including ligand binding and oxygen binding, respectively. As a result, the added value of incorporating protein 3D structures is supported by theoretical foundations in protein property prediction. There are several widely used classification schemes that organize these myriad protein properties, including the Gene Ontology (GO) Consortium [6] and Enzyme Commission (EC) numbers [7]. For instance, Gene Ontology (GO) categorizes proteins into classes within three distinct ontologies—Molecular Function (MF), Biological Process (BP), and Cellular Component (CC)—each representing different aspects of protein properties.

Recent advancements in large pretrained language models (LMs) designed for natural language processing [8,9,10] have inspired the adaptation of such models for protein modeling. In this context, protein sequences are conceptualized as the “language of life,” with AAs serving as the fundamental tokens. Prominent examples of protein language models (PLMs) are primarily transformer-based, such as ESM [11], as well as long short-term memory (LSTM)-based LMs from Alley et al. [12] and Heinzinger et al. [13]. Currently, PLMs have gradually developed toward multimodal large models. Recent studies [14,15,16] have further enhanced their pretrained model capacity by adding structure tokens. Researchers have discovered that language models can capture evolutionary information embedded in billions of protein sequences spanning diverse species. Recent studies [16,17] have shown that language models based on the BERT architecture can predict mutation effects in wild-type sequences. Meanwhile, PLMs are also able to learn fundamental structural information without supervision. For example, protein-level embeddings derived from these models can predict structural classes as defined in SCOPe (Structural Classification of Proteins—Extended) [18]. Additionally, residue-level embeddings have been shown to effectively predict secondary structures and tertiary contact maps [11], even in few-shot learning scenarios [19].

At present, 3D protein structures have also been explicitly incorporated for both PLMs and property prediction tasks. In protein property prediction tasks, 3D structures are predominantly represented as graphs of AAs, which are subsequently processed using graph neural networks (GNNs). Representing 3D structures as amino acid graphs derived from contact maps is a reductionist approach, as it captures only inter-residue distances and interactions while overlooking finer structural details, such as residue orientations. Jing et al. [20] developed a geometric learning method, Geometric Vector Perceptrons (GVP), for learning vector-valued and scalar-valued functions over 3D Euclidean space, with outputs that are equivariant or invariant to rotations and reflections within this space. Zhang et al. [21] further modeled 3D structures by employing different message functions for different edges. Moreover, synergistic prediction methods based on sequence and structure have been gradually developed to predict protein properties. Gligorijević et al. [22] proposed DeepFRI, a GCN-based architecture that integrates sequence and structural information by incorporating amino acid embeddings from protein language models as node features. Wang et al. [23] adopted a similar approach, concatenating language model embeddings with geometric vector perceptron (GVP) node scalar features.

Although existing synergistic prediction methods have achieved certain success in extracting sequence and structural features, these methods typically treat sequence and structure as independent inputs, to some extent neglecting the potential complementarity and synergistic effects between them. For instance, the embedding of sequence features often fails to capture local structural information, while the processing of structural information largely depends on manually engineered feature extraction, which imposes limitations on the current models in terms of multimodal information integration. Currently, pretrained multimodal language models such as ProSST [16] utilize structure quantization based on pretrained GVP and clustering methods to introduce discrete structure tokens. Compared to previous methods, ProSST encodes sequence and structural information as discrete tokens, which are simultaneously used as inputs to the model. It then further refines the relationships between sequence and structure through disentangled attention mechanisms [16,24]. Currently, ProSST has achieved state-of-the-art performance in zero-shot protein mutation prediction. At the same time, previous synergistic prediction methods, which fused sequence and structural information through simple concatenation and weighting, also required a framework for relationship mining of multimodal data and multi-scale information integration to predict protein properties.

Motivated by this development, we designed a novel end-to-end deep learning framework for protein property prediction, integrating information from both protein sequences and 3D structures. We proposed a simple yet effective framework for synergistic prediction based on sequence and structure information, called Sequence-Structure ResNet (SST-ResNet). Building on the representational abilities of multimodal models, we added a multi-scale information integration module specifically tailored for sequence and structure. This module enables comprehensive integration of protein sequence and structure information, thereby maximizing the synergy between sequence and structure in predictive tasks. SST-ResNet was tested on the EC and GO datasets, outperforming prior approaches that relied solely on structural information, sequence information, or their preliminary integration. The source code is available at https://github.com/Zhougv/SST-ResNet/tree/main (accessed on 17 March 2025).

## 2. Results and Discussion

### 2.1. Overview

The SST-ResNet framework is designed for the prediction of protein properties using both sequence and structural information synergistically. As shown in Figure 1, SST-ResNet is divided into two modules: the sequence and structure representation module and the multiscale information integration module. In the first module, the sequence and structure of the protein are simultaneously used as initial inputs, which are processed through a series of encoding steps before being integrated into the multimodal PLM-ProSST. In the sequence input, each amino acid is treated as a sequence token. For structure input, the local structure of each amino acid is considered a structural token. Concretely, the local structure is encoded into a dense vector using a pre-trained structure encoder GVP. Subsequently, a pre-trained k-means clustering model assigns a category label to the local structure based on the encoded vector. Finally, the category label is assigned to the residue as the structure token. ProSST then leverages disentangled attention to fully capture the latent relationships between sequence and structure. The resulting representations are fed into the multiscale integration module, which adopts a ResNet-like architecture [25]. This module employs convolutional kernels of multiple sizes, specifically 3 × 3, 5 × 5, and 7 × 7, to capture hierarchical features at different spatial scales. Each kernel is complemented by batch normalization layers and nonlinear activations, ensuring the stabilization of gradients and enhancing feature expressiveness. Then, the outputs of the multi-kernel convolutions are aggregated and fed into the residual network, facilitating a robust fusion of multi-scale information. Finally, the sequence and structure information, encoded by the multimodal PLM model and multiscale integration, is utilized for the synergistic prediction of protein properties.

### 2.2. Evaluation Metrics

We now provide a detailed explanation of the evaluation metrics for EC and GO prediction. These tasks are designed to address the question of whether a protein possesses specific properties, framing the problem as multiple binary classification tasks. The first evaluation metric, protein-centric maximum F1-score (F_max_), is defined by initially calculating the precision and recall for each protein and then averaging these scores across all proteins. Specifically, for a target protein i, with a decision threshold t∈[0, 1], precision and recall are computed as follows:(1)$${\mathrm{precison}}_{i}(t)=\frac{\sum_{f}1\;[f\in P_{i}(t)\cap T_{i}]}{\sum_{f}1\;[f\in P_{i}(t)]},$$
and(2)$${\mathrm{recall}}_{i}(t)=\frac{\sum_{f}1\;[f\in P_{i}(t)\cap T_{i}]}{\sum_{f}1\;[f\in T_{i}]},$$
where f is a function term in the ontology, $T_{i}$ is a set of experimentally determined function terms for protein i, $P_{i}(t)$ denotes the set of predicted terms for protein i with scores greater than or equal to t, and 1[.]∈{0, 1} is an indicator function that is equal to 1 if the condition is true. Then, the average precision and recall over all proteins at threshold t is defined as:(3)$$\mathrm{precision}(t)=\frac{1}{M(t)}\sum_{i}{\mathrm{precision}}_{i}(t),$$
and(4)$$\mathrm{recall}(t)=\frac{1}{N}\sum_{i}{\mathrm{recall}}_{i}(t),$$
where we use N to denote the number of proteins and M(t) to denote the number of proteins on which at least one prediction was made above threshold $t,\left|P_{i}(t)\right|>0$. Combining these two measures, the maximum F-score is defined as the maximum value of the F-measure over all thresholds. That is:(5)$$F_{\max}=\underset{t}{\max}\{\frac{2.\mathrm{precision}(t).\mathrm{recall}(t)}{\mathrm{precision}(t)+\mathrm{recall}(t)}\},$$

The second metric, the pair-centric area under the precision-recall curve ${\mathrm{AUPR}}_{\mathrm{pair}}$, is defined as the average precision score across all protein-property pairs, which corresponds to the micro-average precision score for multiple binary classification tasks.

### 2.3. Performance of the SST-ResNet

To evaluate the ability of SST-ResNet to predict protein properties, we utilize two challenging datasets, EC and GO. As shown in Table 1, the EC dataset consists of 538 binary classification tasks. The GO dataset is divided into three branches: GO-MF includes 489 binary classification tasks, GO-BP contains 1943 binary classification tasks, and GO-CC comprises 320 classification tasks (see Method for details). We followed the cutoff split methods in Gligorijevic et al. [22] to ensure that the test set only contains PDB chains with a sequence identity of no more than 95% to the training set, as used in Wang et al. [23]. In terms of sample size, the training, validation, and test sets are approximately in a ratio of 8:1:1. The models were subjected to training and performance validation on the training and validation sets to ensure their generalization ability and prediction accuracy. We report two commonly used metrics for evaluating property prediction tasks, Fmax and AUPR. Finally, all the results on the test set are reported based on the highest Fmax score observed on the validation set.

To objectively assess model performance, we systematically compared SST-ResNet with existing frameworks ResNet, Transformer, GVP, GraphQA [26], NewIEConv [27], DeepFRI [22], and LM-GVP [23]. The results are shown in Table 1. For each dataset, we conducted three repeated experiments using different random seeds, while the results of other models were referenced from previous literature [21,28].

As shown in Table 2, SST-ResNet achieved competitive performance on the test set. It obtained the best Fmax scores on the EC, GO-MF, and GO-BP datasets and achieved second-best results on the GO-CC datasets. In Table 2, we also report another popular metric, AUPR. The reported AUPR results are based on the best model selected according to the Fmax metric on the validation set. It can be observed that our method still achieves optimal predictive performance on the EC dataset while attaining the second-best performance on the GO-CC dataset. However, there remains a certain gap between our results and the state-of-the-art performance on other datasets in terms of the AUPR metric. This may be attributed to the inconsistency between the two evaluation metrics. Exploring the relationship between these metrics and developing a model that excels in both would be an interesting direction for future research. In comparison, we find that methods utilizing the synergistic prediction of protein properties based on both sequence and structure generally outperform those relying solely on sequence or structure, which aligns with our previous perspectives.

Enzyme Commission (EC) number prediction aims to identify the EC numbers of various proteins, which characterize their catalytic roles in biochemical reactions. The EC numbers are drawn from the third and fourth levels of the EC tree [29]. SST-ResNet achieved an Fmax score of 0.858 and an AUPR score of 0.81 on the EC dataset, demonstrating a significant advantage over methods relying solely on amino acid sequences or protein 3D structures. Compared to previous synergistic prediction methods integrating sequence and structure, DeepFRI achieved an Fmax of 0.631 and an AUPR of 0.547, while LM-GVP achieved scores of 0.664 and 0.71, respectively. SST-ResNet improved the Fmax score by more than 0.2 and outperformed LM-GVP and DeepFRI in AUPR by 0.1 and over 0.25, respectively.

Gene Ontology (GO) term prediction focuses on determining whether a protein is associated with specific GO terms. These terms categorize proteins into hierarchically related functional classes within three distinct ontologies: molecular function (MF), biological process (BP), and cellular component (CC). SST-ResNet achieves an F_max_ score of 0.613 on the GO-MF dataset, 0.425 on the GO-BP dataset, and 0.463 on the GO-CC dataset. Compared to previous synergistic prediction methods that integrate sequence and structure, SST-ResNet attains the highest Fmax scores on two datasets and the second-best result on one. Its performance on the GO-CC dataset is slightly inferior to that of LM-GVP, while its AUPR metric performance is not particularly outstanding. Overall, these results indicate that our approach more effectively integrates sequence and structure information, fully leveraging their synergy for protein property prediction.

### 2.4. Ablation Experiments

To evaluate the effectiveness of the multi-scale information integration module, we conducted ablation experiments. First, we performed an ablation of the entire module, where the embeddings of ProSST were directly fed into a single-layer MLP head for prediction. Subsequently, we ablated convolutional kernels of different sizes within the multi-scale module, as illustrated in Figure 2. The performance of SST-ResNet is significantly superior to that of ProSST embeddings, indicating the necessity of the scale integration module. Additionally, SST-ResNet consistently outperforms the single-kernel integration module, demonstrating that multi-scale integration of sequence and structural information is also essential. More detailed experimental results can be found in Appendix A. Thus, the incorporation of multiple convolutional kernels enables the extraction of hierarchical features at varying receptive fields, capturing both local and global sequence-structure dependencies more effectively. This comprehensive multi-scale representation enhances the model’s ability to discern intricate biological patterns, enabling the optimal synergistic prediction of protein properties based on sequence and structural information.

### 2.5. Investigation of SST-ResNet Representation

To further validate the effectiveness of SST-ResNet representation, we performed dimensionality reduction and visualization on the hidden layer representations of SST-ResNet. Specifically, we selected the positive samples of the purine ribonucleoside triphosphate binding task (GO:0035639) as the analysis objects. The hidden layer representations of the samples were reduced to 50 dimensions via principal component analysis (PCA) to remove noise, followed by projection to two dimensions using UMAP [30] for visualization. Subsequently, clustering analysis was performed using the DBSCAN [31] method. As shown in Figure 3, the reduced data were divided into eight clusters. We used the Clustal Omega tool (https://www.ebi.ac.uk/jdispatcher/msa/clustalo, accessed on 17 March 2025) [32] to perform multiple sequence alignment (MSA) on a subset of samples from two of the clusters. We observed that samples within the same cluster shared consistent conserved regions, which were confirmed in the MSA. Appendix A documents the detailed experimental setup. By learning both sequence and structural information of proteins, SST-ResNet further reveals their underlying biological properties. In particular, by identifying samples with similar conserved regions, the model generates similar representations. Therefore, we believe that SST-ResNet, through the multi-scale integration of sequence and structural information, can produce high-quality representations for property prediction tasks.

### 2.6. Case Study

ATP (adenosine triphosphate) and rRNA (ribosomal RNA) are two crucial biomolecules in the process of protein synthesis. ATP provides the necessary energy to drive the synthesis and translation of peptide chains, while rRNA is a core component of the ribosome, precisely catalyzing the polymerization of amino acids. The functions of these two molecules are indispensable in protein synthesis and directly influence the life activities and functional execution of the cell.

To further explore the capability of SST-ResNet in learning the latent relationships between protein sequences and structures, we selected two key Gene Ontology Molecular Function (GO-MF) tasks for validation: ATP binding (GO:0005524) and rRNA binding (GO:0019843). First, we applied MMseqs2 to cluster the test set samples, ensuring that sequences were grouped only if their similarity reached at least 40%. Within the same cluster, we selected two sequence-similar samples that exhibited structural and functional label discrepancies.

As shown in Figure 4a, for the ATP binding task, SST-ResNet predicted that the sample protein (PDB ID: 4H1G) has the capability to bind ATP, whereas the sample protein (PDB ID: 4BL8) does not. Similarly, as illustrated in Figure 4b, for the rRNA binding task, SST-ResNet predicted that the sample protein (PDB ID: 51QR) is capable of binding rRNA, while the sample protein (PDB ID: 5MRE) is not. These results demonstrate that SST-ResNet effectively captures the intrinsic relationships between sequence and structure, enabling accurate predictions across multiple critical functional tasks.

## 3. Materials and Methods

### 3.1. Dataset Preparation

In this study, the Enzyme Commission (EC) number [29] and Gene Ontology (GO) term prediction were used to train and evaluate our model. Following DeepFRI [22], the EC numbers are derived from the third and fourth levels of the EC tree, resulting in a total of 538 binary classification tasks. The training set contained 15,550 samples, the validation set comprised 1729 samples, and the test set included 1919 samples. GO terms with a minimum of 50 and a maximum of 5000 training samples were selected. The non-redundant sets were divided into training, validation, and test sets based on sequence identity. All protein chains were retrieved from PDB using the code in their codebase, and obsolete chains with outdated PDB IDs were removed, resulting in slight differences in statistics compared to the original paper. The dataset was divided into a training set with 29,898 samples, a validation set with 3322 samples, and a test set with 3415 samples. The GO dataset was further categorized into three branches: GO-MF, which included 489 binary classification tasks; GO-BP, comprising 1943 binary classification tasks; and GO-CC, containing 320 binary classification tasks. We utilized TorchDrug [33] to download and process the data while filtering out samples with structural issues.The relevant processing scripts can be found at the following link: https://github.com/DeepGraphLearning/torchdrug/blob/master/torchdrug/datasets, accessed on 17 March 2025.

### 3.2. Structure Token Construction

The multimodal PLM ProSST takes protein sequences and structures as inputs. For protein sequences, each amino acid is treated as a sequence token. Additionally, a structure quantization module is required to convert protein structures into discrete structure tokens. In this study, we adopted the structure quantization module and vocabulary from ProSST. The local structure is first encoded into a dense vector using a pre-trained structure encoder GVP. Then, a pre-trained k-means clustering model categorizes the local structure by assigning a category label based on the encoded vector. Finally, this category label is assigned to the residue as its structure token.

We constructed the local structure for each protein, representing the local environment of an amino acid. Centered on a specific amino acid, the nearest amino acids in three-dimensional space were selected. ProSST uses 40 nearest amino acids; however, to reduce computational complexity, we selected the 10 nearest amino acids in this study. Formally, we represented G=(V,E), where V and E denote the residue-level nodes and edges. The edge set $E=\{e_{ij}\}$ includes all i, j for which $v_{j}$ is one of the ten nearest neighbors of $v_{i}$, determined by the distance between their $C_{\alpha }$ atoms. Then, GVP was used as the local structure encoder, the feature extraction function $\pi_{\theta }(G)\in R^{l\times d}$ for the structure. l is the number of nodes, d is the embedding dimension, and θ represents the parameters of GVP. Therefore, for a specified graph G, the encoding process can be described by:(6)$$ST=\frac{1}{l}\sum_{i=1}^{l}\pi_{\theta }(g_{i})$$
where $g_{i}$ represents the features of the local structure associated with the *i*-th node in the graph G and $\pi_{\theta }(g_{i})\in R^{d}$ is the output of the encoder for the *i*-th node. Here, st is the mean pooled output of the encoder and the vectorized representation of the local structure. Notably, to prevent information leakage, the features extracted by GVP do not include amino acid type information and consist entirely of structural cues. In ProSST, GVP is generatively pre-trained on the C.A.T.H dataset [34] by perturbing $C_{\alpha }$ coordinates with 3D Gaussian noise, thereby enhancing its ability to represent structural information. Then, the k-means algorithm was used to identify K centroids within this latent space, denoted as ${\left\{e\right\}}_{i=1}^{K}$. These centroids constitute the structure codebook. Therefore, ST is quantized by the nearest vector $e_{j}$ within the codebook, with j serving as the structure token. K is also referred to as the structure vocabulary size, and we chose K as 2048 in this study.

### 3.3. Multimodal PLM ProSST

To further explore the relationship between protein sequences and structures, thereby achieving better synergistic prediction performance, ProSST employs an expanded form of disentangled attention [24] to integrate the attention of residue sequences, structure sequences, and relative positions. In the subsequent section, single-head attention is used as an example to illustrate the mechanism of sequence-structure disentangled attention in ProSST. Concretely, for a residue at position i in a protein sequence, it can be represented by three items: S denotes its residue token hidden state, ST represents its structure token hidden state, and $P_{ij}$ is the embedding of relative position with the token at position j. The calculation of the cross attention $A_{i,j}$ between residue i and residue j can be decomposed into nine components by:(7)$$\begin{array}{ll} A_{i,j} & =\{S_{i},ST_{i},P_{ij}\}\times {\left\{S_{j},{ST}_{j},P_{ji}\right\}}^{T}\\ & =S_{i}S_{j}^{T}+S_{i}ST_{j}^{T}+S_{i}P_{ij}^{T}+STS_{j}^{T}+STST_{j}^{T}+STP_{ij}^{T}+P_{ij}S_{j}^{T}+P_{ij}ST_{j}^{T}+P_{ij}P_{ij}^{T} \end{array}$$

As described in Equation (7), the attention weight of a residue pair is computed using distinct matrices that account for residue tokens, structure tokens, and relative positions. These matrices facilitate diverse interactions, including residue-to-residue, residue-to-structure, residue-to-position, structure-to-residue, structure-to-structure, structure-to-position, position-to-residue, position-to-structure, and position-to-position. Since the multimodal PLM aims to capture the complex relationships between sequences and structures, certain irrelevant terms are removed, including structure-to-structure, structure-to-position, position-to-structure, and position-to-position. Therefore, the sequence-structure disentangled attention mechanism includes 5 types of attention.

Similar to the standard self-attention operation [19], the computation of the query and key for structure, residue, and relative position, as well as the value for residue, is carried out as follows:$$Q^{s}=SW_{s}^{q}\cdot K^{s}=SW_{s}^{k}\cdot V^{s}=SW_{s}^{v}$$(8)$$Q^{st}=STW_{st}^{q}\cdot K^{st}=STW_{st}^{k}$$$$Q^{p}=PW_{p}^{q}\cdot K^{p}=PW_{p}^{q}$$

So, the attention score ${\hat{A}}_{i,j}$ from residue i to residue j can be calculated as follows:(9)$${\hat{A}}_{i,j}=Q_{i}^{s}{\left(K_{j}^{s}\right)}^{T}+Q_{i}^{s}{\left(K_{j}^{st}\right)}^{T}+Q_{i}^{s}{\left(K_{ij}^{p}\right)}^{T}+K_{j}^{st}{\left(Q_{i}^{s}\right)}^{T}+K_{j}^{p}{\left(Q_{i}^{s}\right)}^{T}$$
where $Q_{i}^{s}$ represents the i-th row of the matrix $Q^{s}$ and $K_{j}^{s}$ denotes the j-th row of $K^{s}$. $K_{j}^{st}$ and $K_{j}^{p}$ are the j-th rows of $K^{st}$ and $K^{p}$, respectively. The term $K_{ij}^{p}$ refers to the row in $K^{p}$ indexed by the relative distance for residue i and j. To normalize the attention scores, a scaling factor of $\frac{1}{\sqrt{5d}}$ is applied to $\hat{A}$. All the ${\hat{A}}_{i,j}$ from the attention matrix are used, and the final output residue hidden state is $H_{0}$:(10)$$H_{0}=\mathrm{softmax}(\frac{\hat{A}}{\sqrt{5d}})V^{s}$$

This is used as the input for the hidden state of the next layer. Notably, ProSST does not mask structure tokens to ensure the correctness of the pretraining strategy.

### 3.4. Multi-Scale Information Integration

After obtaining the output $H\in R^{B\times L\times D}$ from the PLM, we designed a robust multi-scale information integration module. B represents the batch size, L denotes the length of the protein sequence, and D signifies the feature dimension. This module is composed of multiple layers equipped with convolutional kernels of varying dimensions, strategically designed to facilitate the integration of sequence and structural information across diverse scales. By capturing patterns and dependencies at different granularities, this architecture is optimized to enhance the synergistic prediction of protein properties. Based on experimental validation, convolutional kernels with dimensions of 3 × 3, 5 × 5, and 7 × 7 were identified as optimal. These kernel sizes ensure the effective extraction of multi-scale features, thereby contributing to the overall robustness and precision of the predictive framework. We take a convolutional kernel with a size of 3 as an example:(11)$$H_{3\times 3}^{1}=GELU(BN(Conv1D(H,\mathrm{kernel}\_\mathrm{size}=3)))$$
where *GELU* [35] represents the *GELU* activation function and *BN* denotes batch normalization:(12)$$H_{3\times 3}^{2}=BN(Conv1D(H_{3\times 3}^{1},\mathrm{kernel}\_\mathrm{size}=3))$$

Therefore, the output $H^{\prime }$ of each layer in the multi-scale module is:(13)$$H^{\prime }=H_{3\times 3}^{2}+H_{5\times 5}^{2}+H_{7\times 7}^{2}+H$$

Subsequently, an average pooling operation is applied to the output of the final layer to derive a compact and representative embedding of the protein. The mentioned modules work cohesively to enable a thorough and integrative representation by combining sequence information with structural information. Following this integrative process, a simple linear transformation is finally applied for protein property prediction.

### 3.5. Implementation Details

SST-ResNet has been implemented in Python 3.8 and PyTorch 2.0.1 as well as functions in PYG 2.5.2, Scikit-learn 1.1.1, Numpy 1.21.2, Pandas 1.4.3, biotitle 0.39, joblib 1.3.2, and transformers 4.38.2. In the Structure Token Construction module, we selected the 10 nearest amino acids for each amino acid to define its local structure. For the GVP and ProSST models, we retained the default settings provided by the original authors. In the multimodal integration module, the model depth, learning rate, batch size, and hidden layer dimensions were adjusted for each task. For further details, please refer to Appendix A. The specific tuning ranges are detailed in the Appendix A. For the final classification head, a simple linear layer was employed, with its dimensionality determined by the number of classes for each specific task.

### 3.6. Baselines

In this subsection, we describe the implementation details of all baselines.

#### 3.6.1. ResNet

ResNet for protein sequences was proposed by Rao et al. [28]. This model consists of 12 residual blocks and 512 hidden dimensions, and it uses the GELU [35] activation function.

#### 3.6.2. Transformer

The self-attention-based Transformer encoder [8] has demonstrated exceptional performance in natural language processing (NLP). Rao et al. [28] extended the application of this model to protein sequence modeling. This adapted model is comparable in size to BERT-Small [9], featuring 4 Transformer blocks, 512 hidden dimensions, and 8 attention heads, with GELU serving as the activation function.

#### 3.6.3. GVP

The GVP model [20] serves as an effective encoder for protein structures, utilizing iterative updates to refine the scalar and vector representations of a protein. These representations are characterized by their invariance and equivariance properties. Specifically, the GVP model comprises 3 GVP layers, each with 32 feature dimensions, distributed as 20 scalar and 4 vector channels.

#### 3.6.4. GraphQA

GraphQA [26] constructs residue graphs leveraging both bond and spatial information and reimplements the graph neural network. The optimal model consists of 4 layers, featuring 128 node attributes, 32 edge attributes, and 512 global attributes.

#### 3.6.5. NewIEConv

The EC and GO prediction tasks were evaluated using the default hyperparameters reported in the original paper [27], adhering strictly to the standard training procedures for these two tasks.

#### 3.6.6. DeepFRI

DeepFRI [22] is a widely recognized structure-based encoder for protein function prediction. It utilizes an LSTM model to extract residue features and subsequently constructs a residue graph for message propagation among residues, employing a three-layer graph convolutional network [36]. The official model checkpoint was applied for baseline evaluation.

#### 3.6.7. LM-GVP

To further enhance the effectiveness of GVP [20], Wang et al. [23] proposed to prepend a protein language model, i.e., ProtBERT [37], before GVP to additionally utilize protein sequence representations. This hybrid model is also employed as one of our baselines.

## 4. Conclusions

Protein property prediction is a crucial component of drug discovery and precision medicine, playing a vital role in understanding biological functions and the molecular basis of diseases. In this study, we propose a novel protein property prediction framework, SST-ResNet, which integrates a multimodal language model with multi-scale feature fusion to deeply explore the relationship between protein sequences and structures. The model is comprehensively trained on four datasets across EC and GO tasks. Compared to existing methods, SST-ResNet demonstrates superior performance on the test sets, consistently outperforming sequence-based and structure-based encoding models as well as preliminary synergistic prediction models. These results underscore the latest advancements achieved by SST-ResNet in protein property prediction and demonstrate its usability and potential applications in drug design and precision medicine.

There are several promising directions for future work. First, optimizing the construction of protein structure representations is necessary. Methods based on GVP encoders for capturing local protein structures are widely used, yet no systematic evaluation of their effectiveness has been conducted. Second, for more precise protein property prediction, additional information, such as multiple sequence alignment (MSA) data that incorporates evolutionary information, needs to be integrated into the model for learning. Third, enhancing the interpretability of the model is essential. This not only involves accurately predicting protein properties but also providing biochemical insights to assist scientists in experimental validation. Finally, we intend to deploy this method as a web server to facilitate user access and utilization.

## Figures and Tables

**Figure 1 ijms-26-02783-f001:**
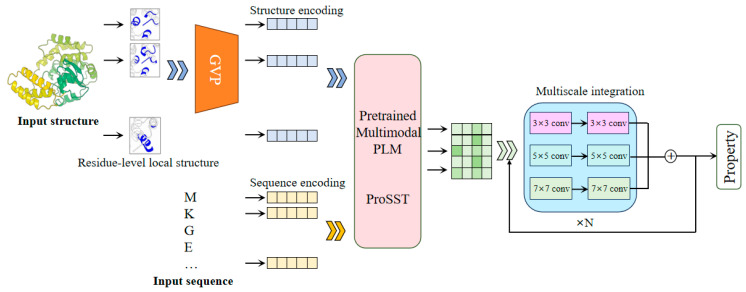
The framework of SST-ResNet. The framework consists of two core components: representation of sequence and structure by multimodal PLM and multiscale integration of information. The sequence and structure information of the protein are simultaneously used as the initial inputs to the model, enabling the synergistic prediction of protein properties through two dedicated modules.

**Figure 2 ijms-26-02783-f002:**
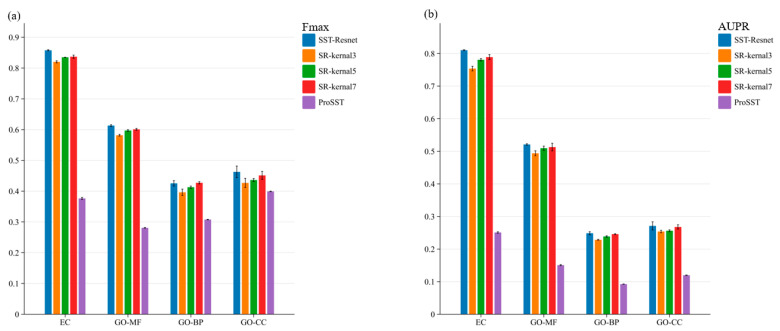
The ablation study of SST-ResNet on the multi-scale information integration module was conducted. Specifically, SR-kernel3 represents the scenario where only convolutional kernels of size 3 were used. ProSST denotes the prediction results obtained using ProSST embeddings. (**a**) Ablation results of the Fmax score for SST-ResNet. (**b**) Ablation results of the AUPR score for SST-ResNet.

**Figure 3 ijms-26-02783-f003:**
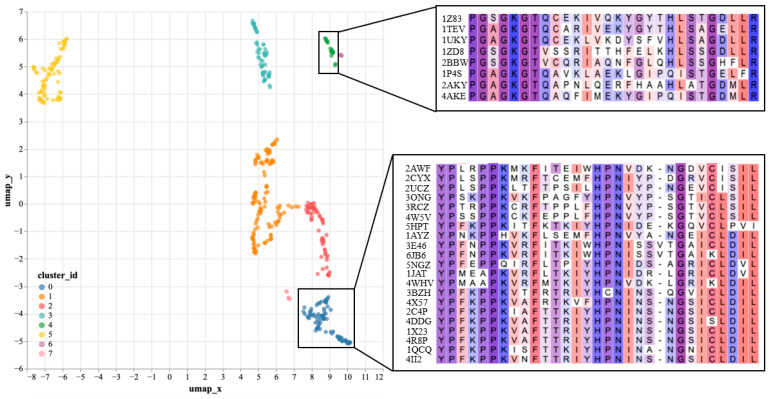
SST-ResNet’s hidden layer representations visualization, including UMAP projection of hidden layer representations and multiple sequence alignment (MSA) analysis of corresponding samples.

**Figure 4 ijms-26-02783-f004:**
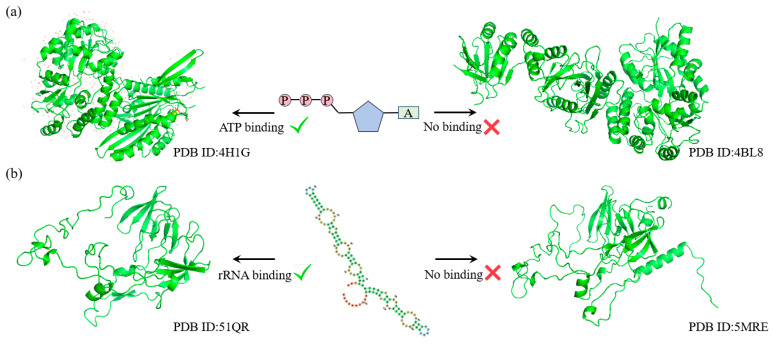
SST-ResNet’s performance on critical tasks, where each set of samples originates from the same sequence cluster but is assigned different labels. (**a**) The prediction results of SST-ResNet on the ATP-binding task for structures 4H1G and 4BL8. (**b**) The prediction results of SST-ResNet on the rRNA-binding task for structures 51QR and 5MRE.

**Table 1 ijms-26-02783-t001:** Task description and data partitioning of the Enzyme Commission (EC) and Gene Ontology (GO) datasets, where the number of tasks represents the number of binary classification tasks.

Dataset	Task	Train	Valid	Test
Enzyme Commission	538	15,550	1729	1919
Gene Ontology Molecular Function	489	29,898	3322	3415
Gene Ontology Biological Process	1943	29,898	3322	3415
Gene Ontology Cellular Component	320	29,898	3322	3415

**Table 2 ijms-26-02783-t002:** Fmax and AUPR on EC and GO prediction. Bold indicates the optimum value in each column. Underscore indicates the suboptimal value for each column.

Method	EC_Fmax_	EC_AUPR_	GO					
			MF_Fmax_	MF_AUPR_	BP_Fmax_	BP_AUPR_	CC_Fmax_	CC_AUPR_
ResNet	0.605	0.59	c	0.434	0.28	0.205	0.304	0.214
Transformer	0.238	0.218	0.211	0.117	0.264	0.156	0.405	0.21
GVP	0.489	0.482	0.426	0.458	0.326	0.224	0.42	0.279
GraphQA	0.509	0.543	0.329	0.347	0.308	0.199	0.413	0.265
NewIEConv	0.735	0.775	0.544	0.572	0.374	0.273	0.444	0.316
DeepFRI	0.631	0.547	0.465	0.462	0.399	0.282	0.46	0.363
LM-GVP	0.664	0.71	0.545	**0.58**	0.417	**0.302**	**0.527**	**0.423**
SST-ResNet	**0.858 ± 0.001**	**0.81 ± 0.001**	**0.613 ± 0.002**	0.521 ± 0.001	**0.425 ± 0.007**	0.249 ± 0.004	0.463 ± 0.02	0.271 ± 0.01

## Data Availability

The datasets and code used in this work are available at https://github.com/Zhougv/SST-ResNet (accessed on 17 March 2025).

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
