# Peer review of "SST-ResNet: A Sequence and Structure Information Integration Model for Protein Property Prediction"

_ijms, 2025, doi:10.3390/ijms26062783_

Round 1

Reviewer 1 Report

Comments and Suggestions for Authors

The manuscript reports the development of a novel end-to-end deep learning framework for protein property prediction, integrating information from both protein sequences and 3D structures. This is based on recently reported pre-trained multimodal language models such as ProSST.

The modern state-of-the-art is well introduced. In the results, the proposed framework, called SST-ResNet, is communicated in a clear manner using the schema and a detailed description of each step of the prediction process. The evaluation metrics (scores) are well explained and the performance framework is evaluated for each method used and the ablation experiments were conducted.

In addition, the ability of SST-ResNet to learn latent relationships between protein sequences and structures was explored on adenosine triphosphate (ATP) and ribosomal RNA (rRNA). The predicted results match the empirical structures of the sample proteins that are either able to bind rRNA or not.

The methods used are well described for each step of the proposed framework.

Reviewer 2 Report

Comments and Suggestions for Authors

This manuscript is well-organized and demonstrates a novel protein property prediction framework, SST-ResNet. This method utilizes protein structural information in addition to protein sequences and outperforms existing platforms. The methods, applications, and prediction results are well-presented and explained. I hope the web server will be available soon.
